# Shibboleth: An agent-based model of signalling mimicry

Jonathan R. Goodman [1,2]*, Andrew Caines[3], Robert A. Foley[1]

1 Leverhulme Centre for Human Evolutionary Studies, University of Cambridge, Cambridge, United Kingdom, 2 Darwin College, University of Cambridge, Cambridge, United Kingdom, 3 ALTA Institute, University of Cambridge, Cambridge, United Kingdom

* jrg74@cam.ac.uk

**Data Availability Statement:** The data underlying the results presented in the study are available at https://github.com/jonathanrgoodman/Shibboleth.

**Funding:** The author(s) received no specific funding for this work.

## Abstract

Mimicry is an essential strategy for exploiting competitors in competitive co-evolutionary relationships. Protection against mimicry may, furthermore, be a driving force in human linguistic diversity: the potential harm caused by failing to detect mimicked group-identity signals may select for high sensitivity to mimicry of honest group members. Here we describe the results of five agent-based models that simulate multi-generational interactions between two groups of individuals: original members of a group with an honest identity signal, and members of an outsider group who mimic that signal, aiming to pass as members of the in-group. The models correspond to the Biblical story of Shibboleth, where a tribe in conflict with another determines tribe affiliation by asking individuals to pronounce the word, 'Shibboleth.' In the story, failure to reproduce the word phonetically resulted in death. Here, we run five different versions of a 'Shibboleth' model: a first, simple version, which evaluates whether a composite variable of mimicry quality and detection quality is a superior predictor to the model's outcome than is cost of detection. The models thereafter evaluate variations on the simple model, incorporating group-level behaviours such as altruistic punishment. Our results suggest that group members' sensitivity to mimicry of the Shibboleth-signal is a better predictor of whether any signal of group identity goes into fixation in the overall population than is the cost of mimicry detection. Thus, the likelihood of being detected as a mimic may be more important than the costs imposed on mimics who are detected. This suggests that theoretical models in biology should place greater emphasis on the likelihood of detection, which does not explicitly entail costs, rather than on the costs to individuals who are detected. From a language learning perspective, the results suggest that admission to group membership through linguistic signals is powered by the ability to imitate and evade detection as an outsider by existing group members.

## Introduction

*And the Gileadites took the passages of Jordan before the Ephraimites: and it was so, that when those Ephraimites which were escaped said, Let me go over; that the men of Gilead said unto him, Art thou an Ephraimite? If he said, Nay;*

**Competing interests:** The authors have declared that no competing interests exist.

*Then said they unto him, Say now Shibboleth: and he said Sibboleth: for he could not frame to pronounce it right. Then they took him, and slew him at the passages of Jordan: and there fell at that time of the Ephraimites forty and two thousand.*

*—Judges 12:5–6*

The Biblical story of Shibboleth, from which the above excerpt derives, describes an extreme situation in which an inability to correctly pronounce a target phoneme results in death. The possibility of this circumstance—where a phoneme's production functions as a password—suggests that, at the very least, members of an in-group are attuned to those who do, or do not, correctly follow linguistic norms [1, 2]. The notion of identifying someone's group identity through pronunciation persists to this day—the differences between Ukrainian and Russian phonology, for example, enables Ukrainians to identify Russian soldiers in the ongoing conflict [3].

Despite the probably apocryphal and exaggerated nature of the Shibboleth story, there is evidence that individuals treat others preferentially based on pronunciation within languages [4–8], even when only two phonetic variables are compared [9]. Previous research has repeatedly demonstrated that individuals across cultures display in-group preferences—that is, people prefer to cooperate with individuals with whom they believe they share similarities ('parochial altruism,' see [10–17]). Social categorization and preferential treatment based on accents has repeatedly been shown in studies in linguistics and psychology [18–22], and even in developmental research in children as young as 5 years [5, 23, 24]. Recent anthropological data even suggest that pronunciation is a greater predictor of preferential treatment than is skin colour in some populations [8].

Research in zoology has also shown evidence of password-like–based preferences in non-human organisms, such as the social sweat bee, *Lasioglossum zephyrum* [25]; see [26] for a discussion. In naked mole rats, furthermore, vocal productions are used as passwords on colony entry: individuals who do not correctly reproduce target sounds are barred from entry [27]. And recent research has suggested that calls among *Spheniscus demersus* are more similar among members of the same colony [28], though the researchers did not explore whether call similarity affected interpersonal preferences.

The relevant evolutionary mechanism that has led to these preferences is unclear. It may be that signals, such as accents, through which cooperative preferences are derived stem from kin selection, where individuals who speak similarly are more likely to share a closer genetic relationship than the average for the population (for discussions, see [14, 29–31]). Similarly, linguistic signals leading to preferential treatment may represent a Greenbeard mechanism [32–35], where phenotypic expression, linked with an allele at a specific locus, is recognizable. Unlike kin recognition, the Greenbeard mechanism does not assume that two individuals have a higher overall genetic relatedness than average in the relevant population.

Models aiming to explain cooperation that do not rely on genetic selection suggest that information about an individual's likely future behaviour guides preferential treatment and cooperation [36–41]. This information may be gained directly, based on how one was treated by the target individual (direct reciprocity; [42]), based on one's observations of how the target individual treats others (indirect reciprocity; [42–47]) or based on information one receives from third parties about the target individual (gossip; [48–54]).

The improved ability to remember and use information about other individuals comes at a substantial cognitive and temporal cost [50, 55–57]. Increasing costs may have selected for categorization of others based on readily observed signals [58, 59]. Using such signals, linguistic or otherwise, individuals would have been able to sort others without the need to resort to

direct or direct information about them. Viewed in this way, social categorization, while obviously error-prone, is a low-cost mechanism for preferential assortment [14, 60].

Broad categorization of individuals based on phenotype or signal, however, leaves open the possibility of exploitation through mimicry [61–63]. Insofar as receivers do not have information, obtained directly or indirectly, about others, free riders will have an opportunity to mimic signals of cooperative intent, or signals of relatedness, that lead to the benefits of cooperation without incurring any costs [14, 64, 65]. The signal may then either become meaningless, or selection may favour receivers that can best detect mimicry.

Previous models have demonstrated how selfish mutants can invade cooperative systems that rely on signalling [62, 66], and further how ethnic markers may evolve from such systems [60, 67]. In these models, selfish strategies undermine the overall group's well-being, and are selected against; individuals who both signal honestly and are able to discriminate between honest and dishonest cooperative signals tend to win out over many generations.

There is, however, an inherent relationship between mimicry and mimicry detection that has not previously been explored in theoretical models, or in empirical research: mimics and detectors are only so effective as their antagonists are ineffective, suggesting that there is an inherent coupling between mimicry quality and the ability to detect mimicry. Moreover, the costs associated with mimicry in previous models are assumed to be low. While we should not assume that, as with the Shibboleth story, a faker is a person who will be killed, there may be a reproductive cost to being discovered as a selfish free rider [68, 69]. In fact, previous research in strong reciprocity presupposes high costs to punished individuals, which range from shame to execution (see [70–73], but see [74] for a critical discussion with commentary). These costs will vary by ecology and society [75], and while we cannot assume they are high, neither should we assume they are low.

In the present paper, we simulate a repeated interaction of individuals from two populations. For ease of explanation, we refer to one population as 'greenbeards' and the second as 'bluebeards', though we do not assume a Greenbeard mechanism is at play—the simulation may also represent a general kin recognition framework, or any in-group affiliation signalling system relying on a hard-to-fake signal (tag-based cooperation, see [76, 77]).

In the simulations of cultural selection, the bluebeard population interacts with the greenbeard population in dyads. We assume, following the Biblical story, that the two groups consist of parochially altruistic individuals. Greenbeards aim to cooperate only with other greenbeards; the bluebeard mimics the Shibboleth signal. The greenbeard's objective is to determine whether the signal is honest, and if they perceive it to be so, they pay a cost to help the bluebeard. Otherwise, they punish the bluebeard (see Methods and Supplement for details).

We work in a scenario where two phonemes occur in allophonic variation, which we represent as 10 colour-variables on an ordinal scale that range from blue to green (see Methods). This is the representation, in our model, of the assumption that there are ten linguistic variables between esh (ʃ; voiceless palato-alveolar fricative; 'green') and s (unvoiced alveolar sibilant, 'blue') that are distinguishable to a listener. We assume that some individuals have green and blue (and everything between) in their cultural inventory; some do not; the individual's mimicry ability in the bluebeard population sets this individually. A receiver's sensitivity or tolerance determines which variables will be accepted as green rather than blue; it is theoretically possible for a receiver to be so poor as to accept totally blue as green, but this is unlikely.

Our aim in five models was to determine whether a single variable that couples mimicry and mimicry detection, $B$, is a better predictor of which cultural traits go into fixation in the overall population (greenbeards and bluebeards) than is the cost of being detected a faker. We explore this in several different circumstances based on the Biblical story of Shibboleth, though we make several assumptions (**S1 Table** in S1 File) and review several variations in the context.

We find overall that detection of mimicry is a better predictor of outcome, as defined by fixation of cultural trait in the population, than is cost of detection. We discuss this in the context of linguistic and evolutionary literature and highlight avenues of future work.

## Methods

### General model description

*(For a full model description according to the ODD standard protocol for agent-based models [78], see the Supplement and **S1 Fig** in S1 File. See **S1 Table** in S1 File for an overview of the models' assumptions and justifications; see **S2 Table** in S1 File for an overview of the evaluated parameters. Visit https://github.com/jonathanrgoodman/Shibboleth for all source code and simulated data).*

Here we describe a general agent-based model that represents two groups of interacting individuals, and suggest that the interactions may lead, over a fixed number of generations, to fixation of a cultural trait in the population. The individuals are each originally a member of one of the two groups, which are labelled greenbeards and bluebeards, respectively. Our notion of a group accords with that of a cultural group, such as a hunter-gatherer band or ethnie (see [14, 79]).

We followed [80] in model design relying on random pairings of simulated individuals without accounting for geography. We assumed instead that the bluebeard group invaded the greenbeard group, and that randomly paired dyads in the two groups interacted as the allophonic traits were selected and reproduced over successive generations. In our models, there were 200 generations for each simulation, which led to a high proportion of runs leading to fixation, but also allowing us to determine the parameter spaces in which any trait going into fixation was less likely.

Individuals in both groups were generated with random *fitness* scores (representing a potential to reproduce [PTR]) ranging from 0–1 and a random *trait* on a predefined trait scale; this represented the likelihood of the individual's allophonic trait being reproduced in the next generation. The scale was any number of the *traits* variable that corresponded to a colour on the blue-to-green spectrum, using the colorRamps package in R [81]. We simulated the model with a *traits* value of 10, which corresponds to the scale of colours shown in Fig 1; again, this represented allophonic traits ranging from esh and s on an ordinary scale. Bluebeards could, on generation 1, have any trait between 1 and 5 (blue, representing s-like sounds); greenbeards could have any trait between 6 and 10 (green, representing esh-like sounds). We assume the traits evolved culturally, rather than genetically, though assume that, within the starting populations, individuals with more similar traits were more likely to be genetically related, following inclusive fitness hypotheses about cooperative signals (see [14]). We also note the analogy with linguistic research into perception of categorical boundaries: there can be a gradience between two related phonemes, with a boundary perceived at some midpoint on the continuum [82].

Greenbeards and bluebeard individuals also had, respectively, *tolerance* and *mimicry* values on initiation. *Tolerance* represented the range of traits an individual was willing to 'accept' on interaction; *mimicry* represented the range of traits an individual could potentially adopt on interaction. Both variables covaried with the globally defined number of *traits* and a second

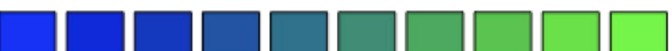

**Fig 1. Scale of traits generated using the colorRamps package when traits = 10.**

global variable, *B* (see below). All individual variables were normally distributed on model initiation.

Aside from the number of individuals per group, *N*, *generations*, and *traits*, we designed two other global variables, *cost* and *B*. *Cost* ranged from 0 to 1 and determined payoffs to individuals upon dyad-interaction, and directly adjusted PTR. *B*, which also ranged from 0 to 1, determined the boundaries of individual *tolerance* and *mimicry* variables along with *traits*; this variable was designed on the understanding that mimicry and detection quality are mirror qualities (see Introduction). The global boundaries for both *tolerance* and *mimicry* were set on initiation using the formula

$$boundaries = (traits - 1) * B \tag{1}$$

where *traits*—1 was rounded to the nearest whole number (We used *traits*-1 rather than *traits* because where B = 1 traits = 10, the latter formula would allow for some individuals accepting traits outside the trait space, leading to an error; see the R code in the online supplement for details). In these simulations, where *traits* = 10, the possible range of *tolerance* and *mimicry* variables were 0 to 9. Where *B* = 0, greenbeards will accept only their own individual traits, and bluebeards will not be able to mimic any traits, and necessarily display only their own. Where *B* = 0.6, greenbeards will tolerate a maximum of half of the total *traits* range, and bluebeards will be able to mimic a maximum half of the total *traits* range. Bluebeards and greenbeards mimic and tolerate, respectively, only traits adjacent on the ordinal scale adjacent to their own. For example, a bluebeard with trait 2 (the second bluest trait in Fig 1) with a mimicry score of 1 will be able to mimic traits 1 and 3, as well as to display its true trait of 2. Similarly, a greenbeard with trait 6 (the bluest green trait on Fig 1) with a tolerance score of 3 will accept trait-signals ranging from 3 to 9 (It is possible that a greenbeard may not accept an extreme green trait (in this scenario, trait 10). This reflects variance in individual receiver ability).

*B* also adjusts global *tolerance* boundaries depending on whether bluebeard-traits make up more or less than half of the total population. Where bluebeards make up less than half of the population, *boundaries* increases the global *tolerance* boundaries range by (*traits*—1) x *B*. This potential to change tolerance boundaries corresponds to research into Batesian mimicry indicating that receivers are more or less tolerant of mimicry depending on whether mimics make up more or less than 50% of the total mimic plus model population [83]. We assume a sharp cut-off of 50% for this adjustment based on the understanding that insofar as mimics invade successfully, receiver sensitivity to mimicry is likely to increase; this represents an element of a competitive coevolutionary relationship between signallers and receivers.

Dyads were paired randomly on each generation and interacted according to the schematic indicated in Fig 2. *Cost* determined payoffs for individuals and was adjusted by an individual's *tolerance* or *mimicry* score. Less tolerant individuals and more tolerant mimics paid a higher cost to reflect the extra fitness costs associated with being a discerning receiver or more precise mimic.

At the end of the dyadic interaction phase, the groups amalgamated into a single population, wherefrom agents reproduced into two new groups with probability *fitness*; this represented trait reproduction in the next cultural generation. Thereafter (ie, from generation 2 onwards), the dyadic interactions were between individuals from the amalgamated population, and any two individuals, regardless of their cultural parent's group, could interact. As the population was mixed after generation 1, whether an agent was a bluebeard or greenbeard was a consequence of which cultural trait they bore, not whether they were a signaller or receiver, which either could be from generation 2 onwards.

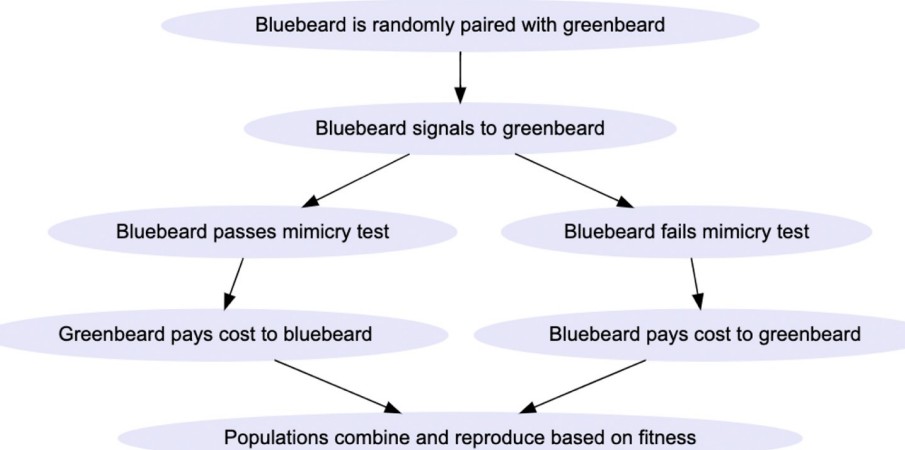

**Fig 2. Flowchart of dyad interactions and fitness effects on individual and global level in population.** Mimicry and tolerance scores are individual-level; cost is global, but coupled for greenbeards and bluebeards only in models 1, 2, and 4. See main text and Supplement for details.

Because of this, *mimicry* was either inherited from a cultural parent or drew randomly from the mimicry boundaries as defined by **Formula** 1 (This was because the population split into two groups every generation, where agents in one group were signal receivers while agents in the second were honest signallers or mimics—representing within-population Shibboleths, rather than the between-group Shibboleth of generation 1. In the receiver group, agents had tolerance scores but not mimicry; in the signaller group, they had mimicry scores but not tolerance. Mimicry was partially heritable because mimicry boundaries did not update; tolerance boundaries updated at each generation, and therefore agents in the receiving group drew randomly from the newly updated tolerance parameter space). Agents were assigned a random new *tolerance* variable based on the newly defined *tolerance* boundaries (note that *tolerance* was also inherited in Model 5). The *fitness* and *trait* variables were inherited without the possibility mutation (This was to maintain simplicity in the model).

## Model 1

We created Model 1, 'simple Shibboleth,' to explore the parameter space fully in a fully agent-based setting (Because of the demanding computational requirements of these models, Model 1 was the only version where we more fully explored the parameter space; in later models we held *cost* and *B*, respectively, at single values while varying the other). In this model, individuals were equally likely to start at generation 1 with any trait (bluebeards, 1 through 5 using the blue2green function in the colorRamps package for R; greenbeards, 6 through 10). The model followed the schematic given in Fig 2.

Greenbeards paid a fitness cost, determined by the global variable *cost*, if the paired bluebeard in the dyad passed the mimicry test; this was adjusted by the greenbeard's *tolerance* score. Bluebeards paid a fitness cost, again determined by the global variable *cost*, to the paired greenbeard if they failed the mimicry test; this was adjusted by the bluebeard's *mimicry* score (In the case of greenbeards, cost was adjusted as a function of the number of traits and the individual's tolerance score with the formula (cost + (number of traits—tolerance)); in the case of bluebeards, cost was adjusted at the individual level using the formula cost + mimicry. See the online supplemental material for all source code). All costs were therefore symmetrical in

the randomly paired dyads and in the population overall; this reflected the circumstances given in the Shibboleth story as outlined in the Introduction. We ran the model using cost = 0.1 and varying B from 0 to 1 by units of 0.1, and then varied cost from 0 to 1 while holding B at 0.6 (B was held at 0.6 rather than 0.5 as the former was the value at which neither trait-group had an advantage in mimicry vs mimicry detection (see Results). Computational power prevented our running these models in the full parameter space; each 500-run iteration where generations = 200 and N = 50 took ~6 hours on a high–processing power computer. We capped generations at 200 as majority of runs went into fixation by this point but nonetheless allowed for analysis of which model values made fixation less likely). We then ran the model varying cost using different values of B (B = 0, 0.25, 0.75, and 1 in addition to 0.6) to explore whether cost was likely to predict outcome anywhere in the parameter space. Finally, we ran the model varying B using different values of cost (0, 25, 50, 75, and 100 in addition to 10).

## Model 2

Model 2, 'cost decoupling,' explored whether creating an asymmetry between costs paid by greenbeards and bluebeards affected which traits went into fixation by generation 200. In this model, the operations were identical to Model 1, except while bluebeards paid cost adjusted by individual mimicry on failing the mimicry test, greenbeards paid only the adjusted cost of their tolerance score on failing to catch a mimic. This reflected a circumstance where greenbeards did not provide costly help to bluebeards where bluebeard passed a mimicry test. We ran this model varying B where cost = 10 and varying cost where B = 0.7 (The value where B did not predict outcome; see Results).

## Model 3

In Model 3, 'population cost,' we explore the effects of adding in a population-level effect to all interactions, combining agent-based modelling and population modelling. This model is identical to simple Shibboleth except, where a bluebeard passed a mimicry test, all bluebeards gained fitness equal to cost / 10 and all greenbeards lost fitness equal to cost / 10; these fitness effects were inverted in the case of a bluebeard failing a mimicry test. These effects were motivated by the idea that a bluebeard being admitted into the greenbeard population introduced a wider pool of cultural traits and thus a loss of PTR for green cultural traits (ie, traits 6–10 in Fig 1). Further, a bluebeard being detected as a mimic raises the likelihood that future mimics will be detected (ie, as knowledge of mimicry by bluebeards spreads in the greenbeard population). As with model 2, we explored which traits went into fixation with generations = 200 and N = 50 while varying B and cost, respectively.

## Model 4

In Model 4, 'blue-most,' greenbeards had a five times higher chance of adopting traits 6 and 7 (that is, the green traits closest to blue on in Fig 1), which reflected a circumstance where, in the absence of between-group competition, functional selection [84] might drive the 'esh' phoneme used in greenbeard populations to be less extreme (ie, more like 's'). Furthermore, the greenbeard group paid twice as high a population-level cost when any bluebeard passed a mimicry test (cost / 5 for the whole greenbeard group) (Five kept the majority of greenbeards with traits 6 and 7 with a minority (~5 where N = 50) of stricter greenbeards with traits 8–10. Again, this reflected the notion that where there is not a strong cultural selection pressure, cultural traits (specifically, phonemes) are likely to be weaker signals (for a detailed discussion, see [84] on functional selection). Greenbeards play twice the value of cost to emphasize the importance of within-group enforcement mechanisms as according to cultural group selection

theory (see, for example, [12]); we reduced tolerance to 0 to indicate that, again following cultural group selection theory, a subset of in-group members are likely to pay a significant cost to maintain within-group homogeneity in cultural traits). We developed this model to evaluate whether, when greenbeards disproportionately adopt the two 'blue-most' traits at generation 1, bluebeards will be advantaged with the same variables as in Model 3.

### Model 5

Lastly, in Model 5 ('punisher'), we also introduced a new mechanism, altruistic punishment [72, 85], to determine the effects of within-group sanctioning mechanisms in the evolutionary literature in our outputs. Here, we removed the function through which tolerance boundaries updated at each generation. Instead, heritable individual *tolerance* scores were adjusted socially. In the case of a high-tolerance greenbeard accepting a bluebeard, a low-tolerance greenbeard paid a fitness cost to reduce the high-tolerance greenbeard's *tolerance* to 0. This cost was twice the *cost* variable. The model was otherwise identical to Model 4.

### Analyses

We first ran an unbiased transmission model (see [80, 86]) 5000 times. This model had no selection or signalling system and, consequently, we assumed that, on each run, which trait went into fixation was a matter only of chance. This served as a control data set with the results from the four models to determine whether selection, as coded in our models, affected outputs.

We ran each Shibboleth model for 500 simulations per parameter assignment; all 500 simulations were averaged after completion. We created binomial and multinomial general linear models to compare results between the data sets generated. All modelling was conducted in R [87]; results were represented graphically using the ggplot2 [88] package for R; all analyses were conducted using R base functions and the nnet package [89]. See the supplementary material and S2 Table in S1 File for further information about the parameters used.

Our aim is to analyse, using these five model variations, the relevant importance of risk of detection and cost of detection in an intergroup conflict where a signal is used to determine affiliation.

## Results

### Unbiased transmission (drift)

Fig 3 and **S3 Fig in** S1 File show, respectively, the breakdown of results from 5000 runs, and a typical sample run, of the drift model without a biased transmission process. As expected, green traits went into fixation—that is, the entire population adopted a single trait—at a similar rate as did blue traits (2353 vs 2450, respectively). The median generation at trait fixation was 53, and 197 runs had no trait at fixation at generation 200.

### Model 1—B variance

We initially held *cost* steady at 10 and the *B* variable from 0 to 1 in increments of 0.1. We then averaged the 10500 runs into a single data frame, using the 5000 drift runs as a reference. Overall, not including drift runs, a green trait went into fixation in 2317 of runs, compared with 2085 runs for blue traits (1098 unresolved). Compared with the drift model, adding signalling and selection mechanisms significantly reduced the likelihood that any trait would go into fixation by generation 200; this was true across the *B*-parameter space (see online supplementary material).

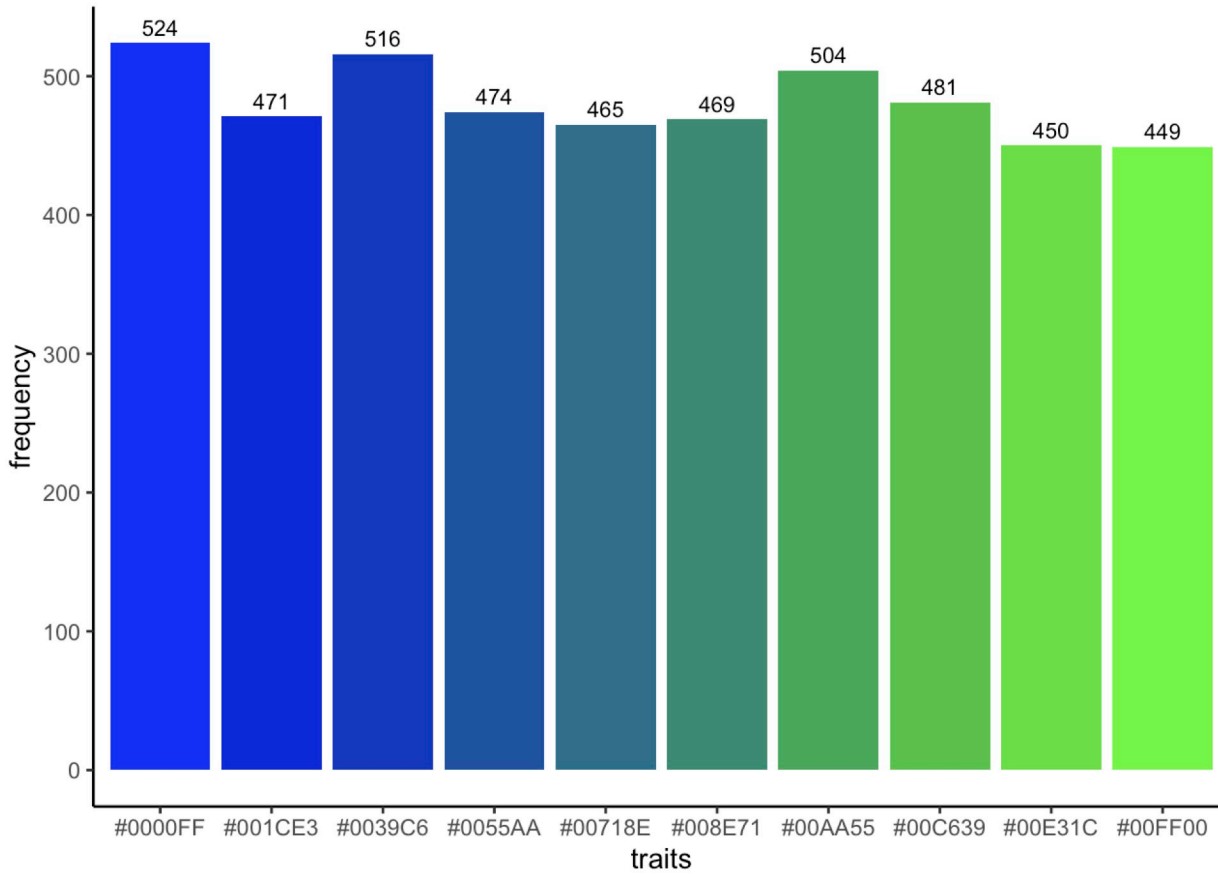

**Fig 3. Traits at fixation by final generation in the drift model.** x-axis = trait on blue-green spectrum; y-axis = number of times trait went into fixation over 5000 runs of drift model; outcomes were distributed nearly evenly across evaluated traits. Parameters: N = 50; traits = 10; generations = 5000.

Fig 4 shows, at generation 200, the average trait frequency distribution as *B* increases from 0 to 1. Fig 5 gives the odds ratios (ORs) for a green outcome for increments of *B*, as compared with the results from the drift-only model. The *B* variable is a strong predictor of which trait-group goes into fixation, even with a low cost of 10. The traits are equally distributed when *B* = 0.6. See the supplementary material for a graphical representation of a sample run (S3 Fig in S1 File).

Finally, we ran further sets of the model, changing the value of *cost* for each set while varying *B*. We used *cost* values of 0, 10 (reusing results from the above), 25, 50, 75, and 100; all other parameters were identical to the previously described runs. In each set of runs except where *cost* = 0, *B* continues to predict outcomes in a negatively correlated manner (Fig 6). Where *cost* = 0, green outcomes are predicted only where *B* = 0.1 (OR = 1.40; $P < .01$) and *B* = 0.2 (OR, 1.55; $P < .001$). Where *B* > 0.5, a blue outcome was significantly more likely (see online supplemental material).

## Model 1—Cost-variance

We next held *B* at 0.6, as this was the only value at which *B* did not predict an outcome (OR = 1.00; Fig 6), and varied *cost* from 0 to 100 in increments of 10. We again averaged the

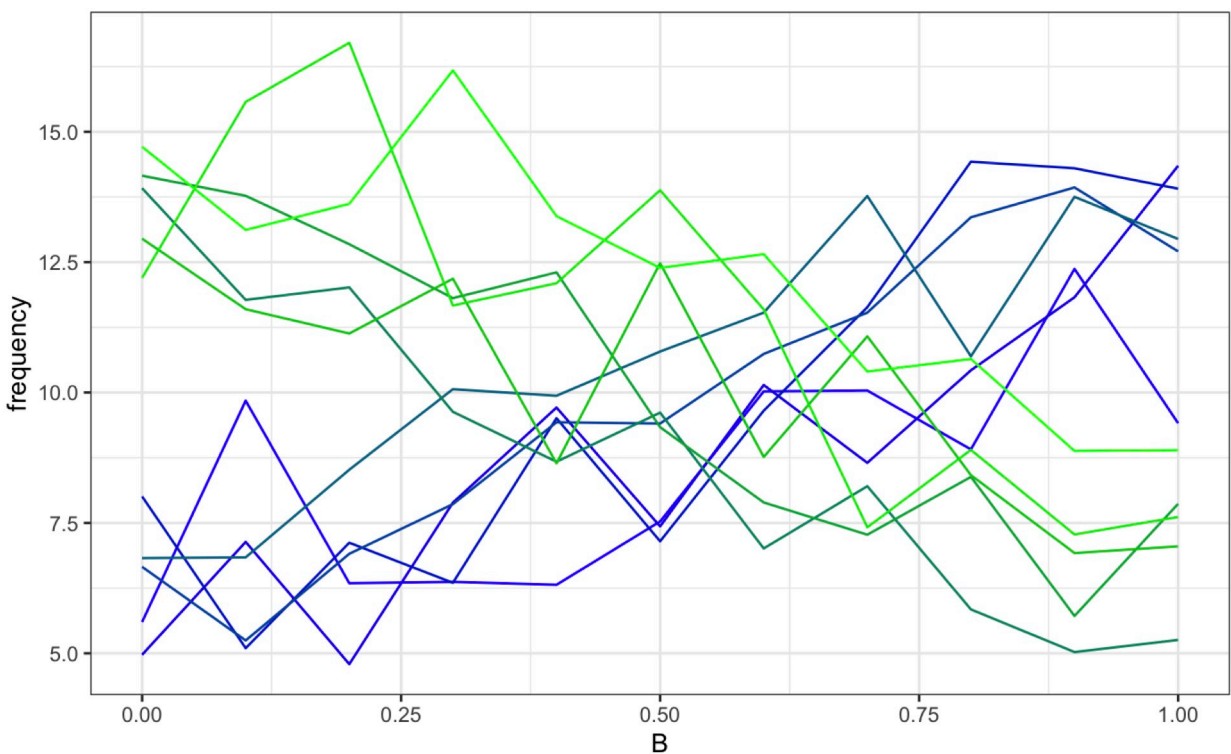

**Fig 4. Average trait (of 10 traits on blue-green spectrum) frequency in model 1 (simple Shibboleth).** X-axis = value of B, increasing from 0 to 1 in increments of 0.1; 500 runs per value; y-axis = average frequency of each trait at final generation in a population of 100 individuals (N = 100). The frequency of individuals with a green trait at final generation drops with each 0.1 increase of B.

10500 runs into a single data frame. Overall, not including the drift runs, a green trait went into fixation in 2164 of runs, compared with 2248 runs for blue traits (1088 unresolved). The median generation at which any trait went into fixation was 82. As where we varied *B*, each variable for *cost* had a significantly lower OR for any trait going into fixation at generation 200, compared with drift (**S4 Fig in** S1 File).

Fig 7 shows, at generation 200, the average trait frequency distribution as *cost* increases from 0 to 100. *Cost* is a poor predictor of which trait-group goes into fixation, a finding that we discuss below.

As when we varied *B*, we ran further sets of the model, changing the value of *B* for each set while varying *cost*. We used *B* values of 0, 0.25, 0.6 (reusing results from the above), 0.75, and 1; all other parameters were identical to the previously described runs. In each set of runs except where *cost* = 0, the static *B* value appeared to predict outcomes in a negatively correlated manner, though *cost* did not (Fig 8). See the online supplemental material for the OR for a green outcome for each combination of *B* and *cost*.

**Discussion.** Analysis of runs from model 1 show that *B* predicts, across values of *cost*, whether a green or blue cultural trait will go into fixation by generation 200. The exception is where *cost* = 0, in which case varying *B* did not show consistent results: where *B* = 0, neither a green trait nor a blue trait was more likely to go into fixation, though where *B* = 0.1 or 0.2, green traits were significantly more like (ORs, 1.40 and 1.55; both *P* < .01; see online supplemental material). For all values greater than *B* = 0.5, a blue trait was more likely, following trends from greater values of *cost*.

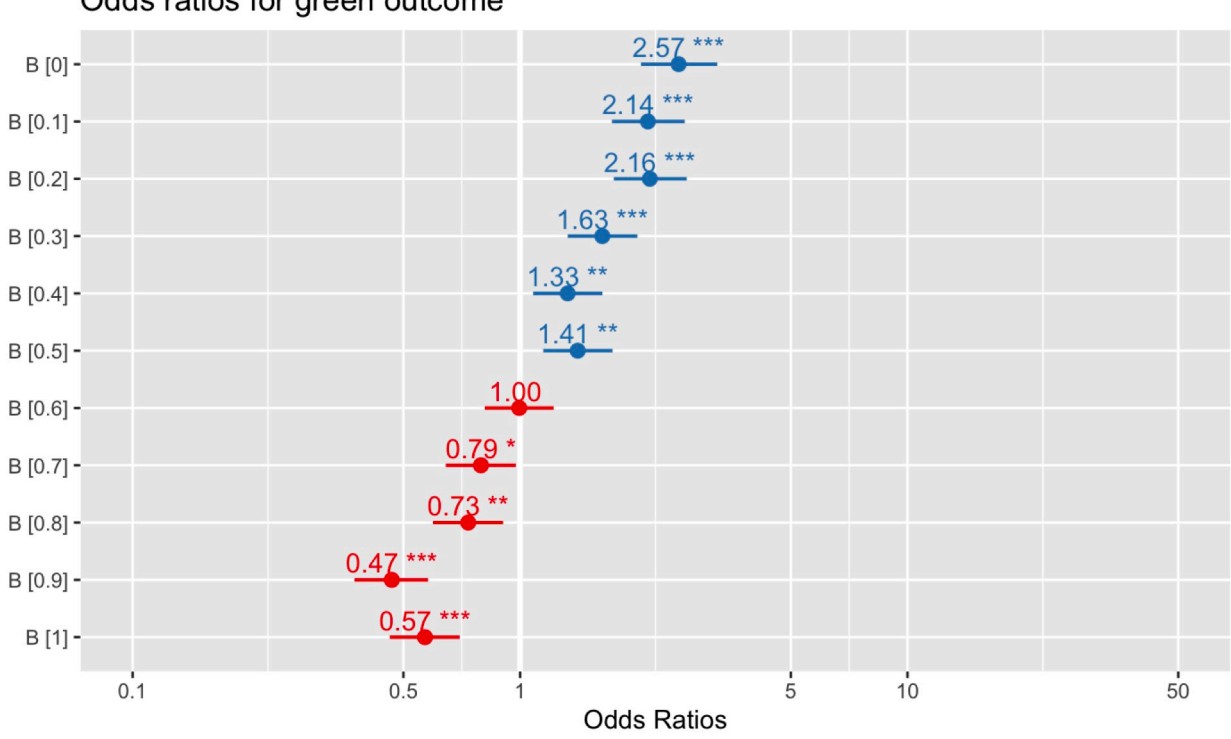

**Fig 5. Odds ratios (ORs) for a green outcome (ie, any green trait going into fixation), compared with drift, as B increases from 0 to 1 in Model 1.** *** indicates P < .001; ** indicates P < .01; * indicates P < .05; cost = 10.

Interestingly, increasing the value of *cost* did not dramatically increase the odds that a green trait would go into fixation at low values of *B*. Where *B* = 0 and *cost* = 100, theoretically the point at which the OR for a green outcome should be highest, the OR was only 2.58 (*P* < .001), a nearly identical finding to where *cost* = 10 and *B* = 0 (OR, 2.57; *P* < .001), indicating that increasing *cost*, even at low values of *B*, may not affect outcomes.

## Model 2—Cost decoupling

In Model 2, the operations were identical to Model 1, except greenbeards paid only the adjusted cost of their tolerance score on failing to catch a mimic. All runs across Model 2's parameter space were less likely to show any trait going into fixation than with the drift comparator model.

When we held *cost* steady at 10 and varied *B*, as with Model 1, the frequency of green traits in the population dropped at generation = 200 as *B* increased from 0 to 1 (**S5 Fig in** S1 File). The design of Model 2, unlike with Model 1, showed that green traits were favoured through *B* values of up to 0.7 (0.6 in Model 1), showing the updated cost decoupling slightly favoured green cultural traits, though even where *B* = 0, the OR for a green outcome was only 2.11 (*P* < .001; **S6 Fig in** S1 File).

When we consequently held *B* at 0.7 (We used this value, rather than 0.6, as it was the only value of *B* for which neither green nor blue traits were more likely to go into fixation). and varied *cost*, a *cost* value of 0 showed that blue traits were more likely to go into fixation (OR for a green trait, 0.68; *P* < .001; **S7 Fig in** S1 File), though higher values showed either no significant finding or a favour for green outcomes (notably where *cost* = 50 and 100; **S7 Fig in** S1 File).

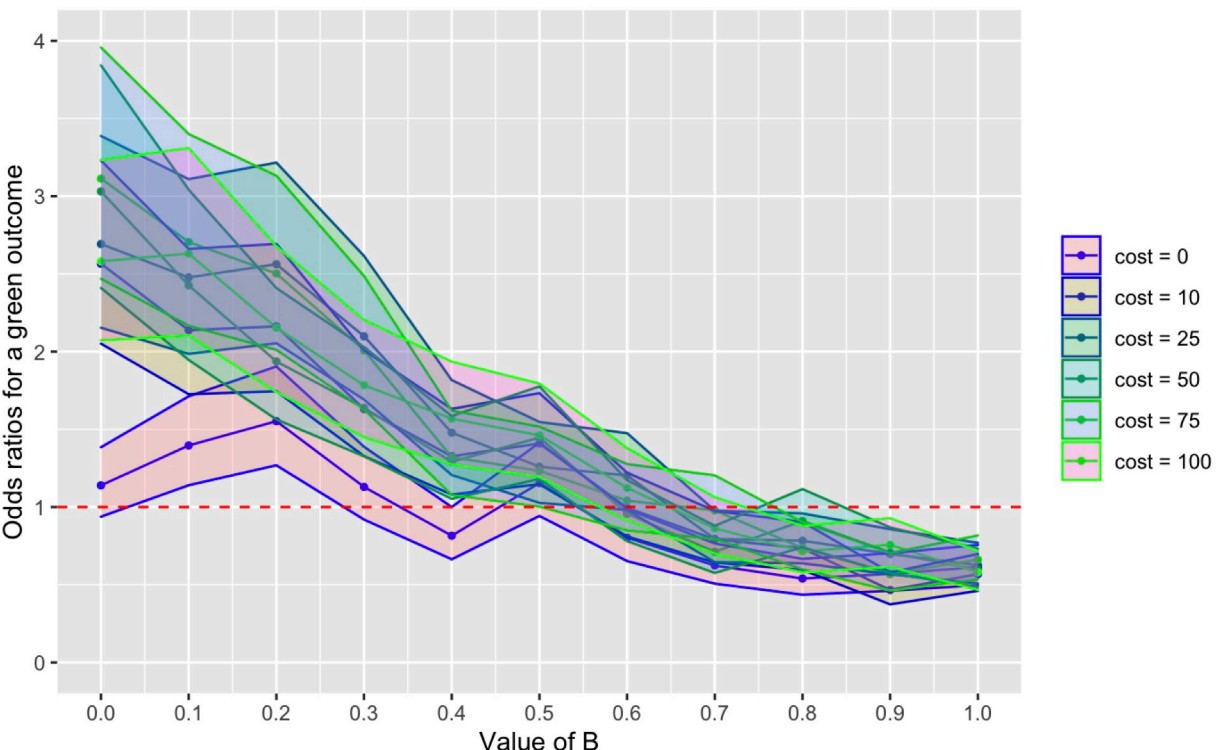

**Fig 6. Odds ratios (ORs) for a green outcome (ie, any green trait going into fixation) with confidence intervals (shaded areas), compared with drift, as B increases from 0 to 1, for runs where we used different steady values of cost.** We used cost values of 0, 10 (reusing results from the above), 25, 50, 75, and 100; all other parameters were identical to the previously described runs. In each set of runs except where cost = 0, B continues to predict outcomes in a negatively correlated manner. Where cost = 0, green outcomes are predicted only where B = 0.1 (OR = 1.40; P < .01) and B = 0.2 (OR, 1.55; P < .001). Where B > 0.5, a blue outcome was significantly more likely (see online supplemental material). *** indicates P < .001; ** indicates P < .01; * indicates P < .05.

**Discussion.** As expected, decoupling costs paid in the greenbeard vs bluebeard groups increased the proportion of the *B*-parameter space in which green traits went into fixation. The degree was, however, less pronounced than expected, and low levels of *B* did not favour green traits to a greater degree than that seen in Model 1. This shows that only requiring that greenbeards pay costs related to their own tolerance score and not the global *cost* variable value did not drastically alter outcomes from where costs were not decoupled.

We did note, however, that where costs were recoupled in this model—that is, where *cost* = 0, both greenbeards and bluebeards paid costs of only their respective tolerance and mimicry scores—blue traits were favoured where *B* = 0.7. As *cost* increased in this case, green traits were favoured, though this was not consistent across the *B* parameter space (**S7 Fig in** S1 File).

## Model 3—Population cost

In Model 3, we removed the cost decoupling element of Model 2 and added a population-level cost in addition to the costs used in Model 1 (see Methods). As with Model 1, all runs showed that, compared with the drift model, any trait going into fixation was less likely (see online supplemental material).

When we held *cost* at 10 and varied *B*, we found that adding the population-level effects drastically increased the effect the value of *B* had on the frequency of green vs blue traits at

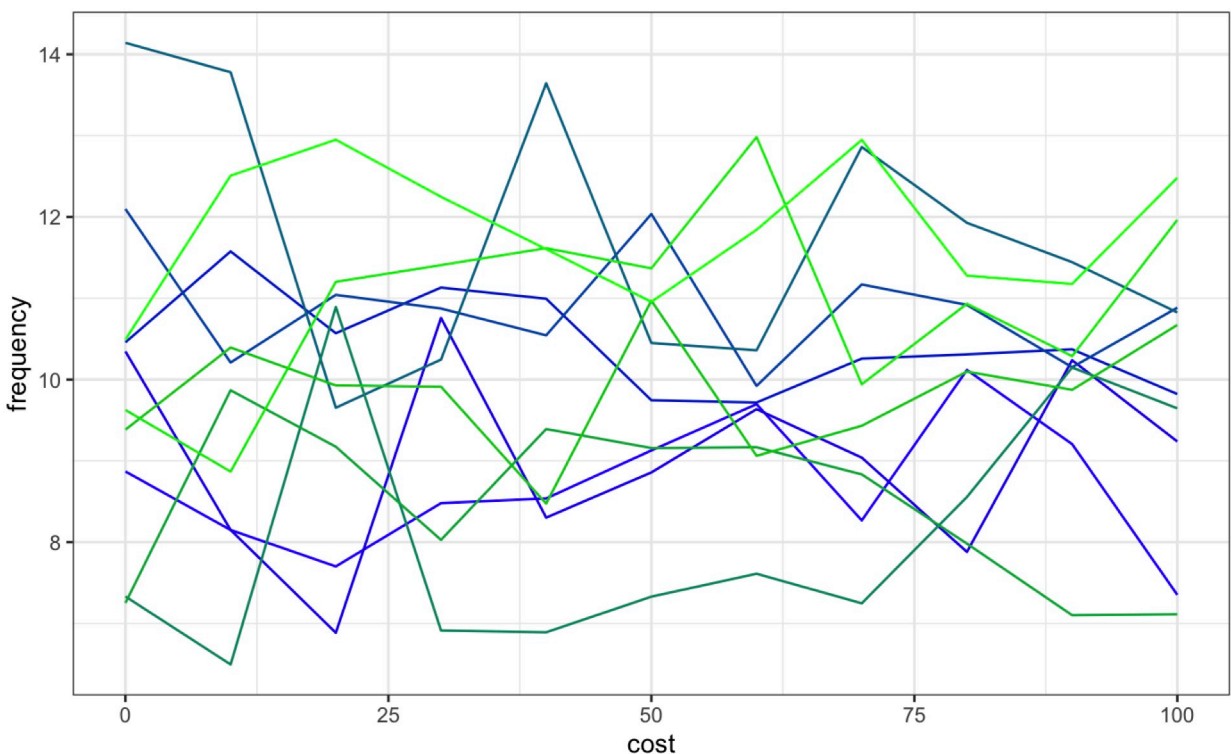

**Fig 7. Average trait (of 10 traits on blue-green spectrum) frequency in model 1 (Simple Shibboleth).** x-axis = value of cost, increasing from 0 to 100 in increments of 10; 500 runs per value; y-axis = average frequency of each trait at final generation in a population of 100 individuals (N = 50). Varying cost does not affect which trait goes into fixation where B = 0.6.

generation 200 (Figs 9 and 10a) as well as the OR for a green outcome (**S8 Fig in** S1 File). Where $B = 0$, the OR for a green outcome was 110.63 ($P < .001$); where $B = 1$, this value was 0.26 ($P < .001$). Varying *cost* while holding $B$ at 0.6 did not appear to predict outcomes (supplemental material and **S9 Fig in** S1 File and Fig 10b).

**Discussion.** The results from Model 3 are similar to those of Model 1, except adding in population-level effects drastically increases the predictive power of the *B* variable across the *cost* parameter space. Even where costs were low, a *B* value of 0 increased the odds of a green outcome, compared with drift, by more than 100-fold. The purely agent-based model showed odds of less than 3 with the same value, indicating that adding a low population-level cost can have profound effects on the cultural transmission process. Again, however, *cost* did not appear to be a strong predictor of outcomes.

## Model 4—'Blue-most'

In model 4, green agents were 5 times more likely to adopt the two 'bluest' traits on the spectrum. Across this parameter space, any trait was less likely to go into fixation than with the drift model (see online supplemental material). When we held *cost* steady at 10 and varied the *B* variable, we found again that the *B* variable is a strong predictor of which trait-group goes into fixation; the traits are, again, equally distributed when $B = 0.6$. The OR for a green outcome when $B = 0$ is, however, less than half that found in Model 3 (50.09 vs 110.63, respectively; see online supplemental material and Fig 10a). When we held *B* steady at 0.6 and varied

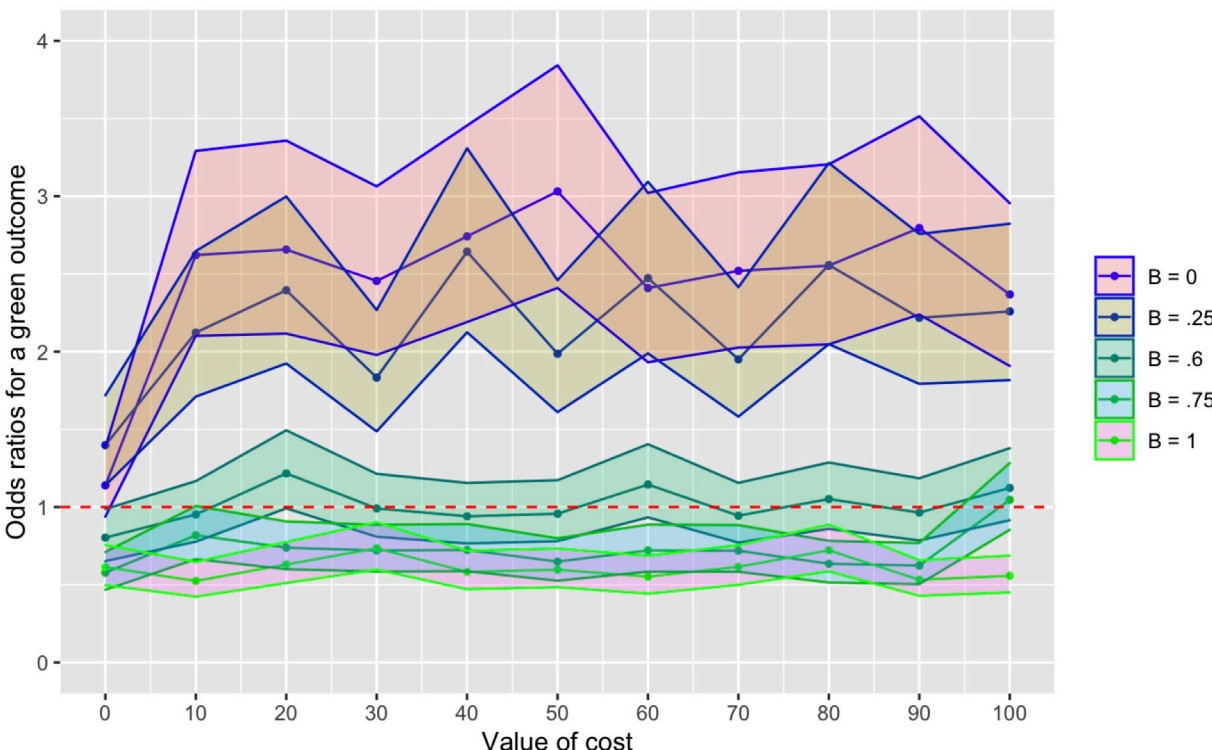

**Fig 8. Odds ratios (ORs) for a green outcome (ie, any green trait going into fixation) with confidence intervals (shaded areas), compared with drift, as cost increases from 0 to 100, for runs where we used different steady values of B.** We used B values of 0, 0.25, 0.6 (reusing results from the above), 0.75, and 1; all other parameters were identical to the previously described runs. In each set of runs except where cost = 0, the static B value appeared to predict outcomes in a negatively correlated manner, though cost did not. See the online supplemental material for the OR for a green outcome for each combination of B and cost.

*cost* from 0 to 100, outcomes were not strongly predicted (see online supplemental material and Fig 10b).

**Discussion.** Where *cost* was held at 10, the 2 blue-most green traits on the blue-green spectrum were disproportionately more likely to go into fixation where $B \leq 0.5$ (**S10 Fig in** S1 File). While the odds ratio for any green trait going into fixation was drastically higher where $B = 0$ than with any other value, the variance was lower in the range of values favouring a green trait (1.86 to 50.09). This accords with our prediction that, where more green agents adopt a blue-most trait at generation 1, the model will have fewer cases of green traits going into fixation. Notably, however, this pattern of blue trait benefits starts to disappear at values of $B > 0.2$.

Where *B* was held at 0.6 while cost was varied, the pattern was similar to that found in the population-effects model (Model 3). *Cost* values of 0 and 30 were weakly predictive ($P < .05$ and $> .01$) of a blue trait going into fixation, suggesting there may be a small benefit to mimics where *cost* is minimal and sensitivity to mimicry is low—but this finding may need further exploration in another iteration of this model (see online supplemental material).

## Model 5—Altruistic punishment

In model 5, green agents were both 5 times more likely to adopt the two 'bluest' traits on the spectrum and paid a fitness cost to punish other greenbeards who did not have sufficient

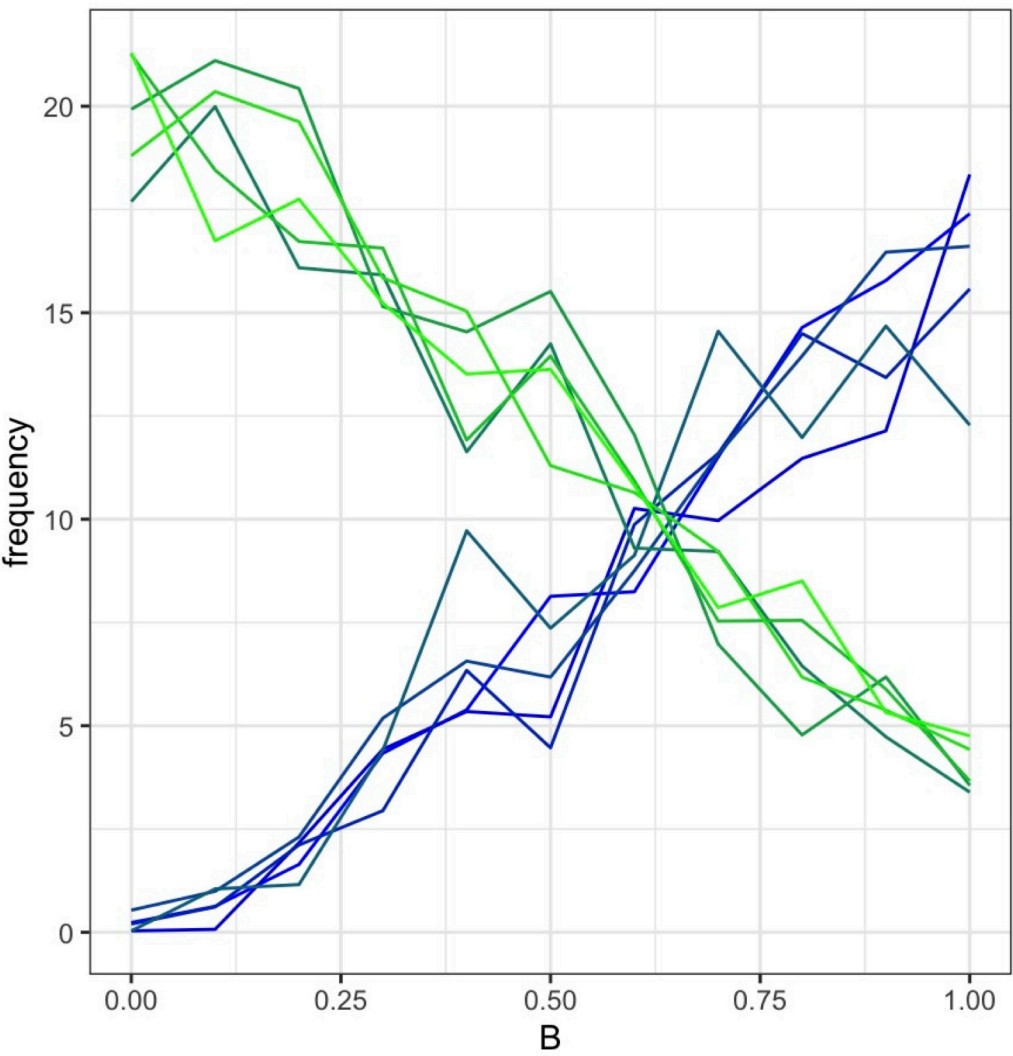

**Fig 9. Average trait (of 10 traits on blue-green spectrum) frequency in Model 3.** x-axis = value of B, increasing from 0 to 1 in increments of 0.1; 500 runs per value; y-axis = average frequency of each trait at final generation in a population of 100 individuals. The frequency of individuals with a green trait at final generation drops with each 0.1 increase of B. We held cost steady at 10.

sensitivity to mimicry. When we held *cost* steady at 10 and varied the *B* variable, green outcomes (any green trait going into fixation) were more likely from $B = 0$ through to 0.7 (**S11 Fig in** S1 File). Compared with Models 3 and 4, however, the odds were less pronounced, with an OR for a green outcome of 6.18 ($P < .001$) where $B = 0$ (Fig 10a). For this model, we also held *B* at 0.8 (the value at which the OR for outcome was most centred around 1; **S11 Fig in** S1 File) and varied cost. Varying cost to any value other than 0 or 10 predicted a green outcome (**S12 Fig in** S1 File).

**Discussion.** The punisher model combined the blue-most elements from model 4 with an altruistic punishment mechanism (akin to that described by [90]; see also [91]), where a highly sensitive greenbeard paid a cost to increase a low-sensitivity greenbeard's sensitivity to mimicry when a mimic 'fooled' the latter. Consistent with predictions from theorists in the strong

(a)

(b)

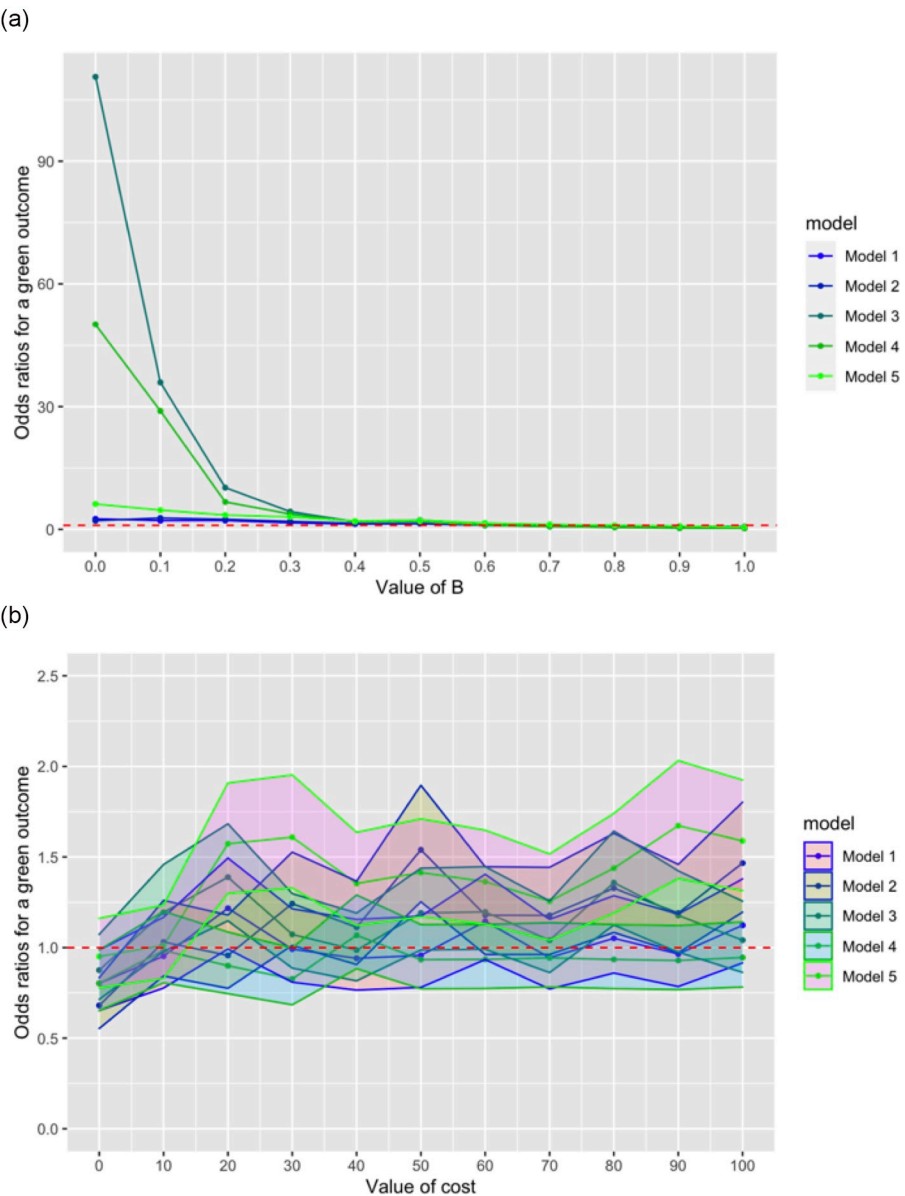

**Fig 10. a (top).** ORs, across all 5 models, for a green outcome (y-axis) as B increases from 0 to 1 (no CIs used to maintain viewability; see **S13** and **S14 Figs** *in* S1 File for further plots where CIs are visible); cost = 10 in all cases. While we investigated Model 1 the most closely, we note that all models with population-level effects (Models 3–5) show increased effects of varying B compared with no population-level effects (Models 1 and 2). **b (bottom).** ORs with 95% CIs, across all 5 models, for a green outcome (y-axis) as cost increases from 0 to 100; in all models, we used the variable for B where neither outcome is more likely. Cost is not a strong predictor of outcome across all models. Parameters for all models: generations = 200; N = 50; runs = 500. 5000 drift runs were used as a comparison.

reciprocity literature [72], the model favoured greenbeards in a greater range of *B* values (including 0.6, the halfway mark where significance was not reached either way in Models 1 and 2). An unexpected finding, however, was that the values of *B* that were favoured in Models 3 and 4 ($B \leq 0.5$) had much lower odds ratios favouring green traits than those seen in the previous models, with $B = 0$ showing an OR of only 6.18, drastically lower than the 110.63 in Model 3.

Yet when we varied *cost* and held *B* steady at 0.8, all values greater than 10 favoured a green-beard outcome—showing that, where a punishment mechanism is implemented, higher costs may favour the punishing group that reduces within-group behavioural heterogeneity [92–94]. In our model, this is, however, contingent on the punishment mechanism improving, rather than reducing, the receiver's fitness, according with a "do as we do, and you will benefit" system (this may accord with alternative subsets of any basic punishment or policing mechanism, see [85] for discussion).

## General discussion

Our central finding, over these 5 models, is that a sensitivity/mimicry spectrum–variable is a strong predictor of which of 2 groups of cultural traits in competing populations will go into fixation on repeated interaction (Fig 10a). Cost, which reflects both cost to the honest signallers (greenbeards) when fooled, and cost to mimics (bluebeards) when caught attempting to fool, is, however, a poor predictor of outcomes in Models 1–4 (Fig 10b). This conflicts with the perhaps intuitive view that greater costs necessarily imply a greater likelihood of a desired outcome individuals in a group may have: when faced with sorting individuals by a desired quality, a greater risk of paying high cost may not lead to a greater likelihood of success by the sorter. Put generally: to maintain in-group homogeneity, it is better, according to these models, to be sensitive to invasion by outsiders, even where costs of invading are low, than to have high costs but a low rate of invader detection. This accords with laboratory findings suggesting that individuals are less likely to cheat in a game when the perceived risk of being caught, rather than the sanctions against those who are caught, is greater [95, 96], though this finding may be culturally dependent.

Findings from the drift model echo the 'r-shaped' effects found in cultural evolution models (see [97, 98]). Shortly after the model initiates, one trait shows a much higher chance of being copied only because more individuals, by generation 2, have that trait than any other. The model then quickly reaches fixation, with a median number of generations to trait-fixation of 53. Only a small number of runs (3.9%; 197 of 5000) of the drift model did not reach fixation at all.

In our models, we found that *B*, the variable that determined the global boundaries of mimicry and tolerance scores, was a strong predictor of whether a green or blue trait went into fixation. At any iteration of the model where *cost* = 10 and $B \leq .5$, the odds ratio (OR) for a green trait going into fixation was at least 1.87, though as high as 110.63 where *B* = 0. Models 1 and 2, which were explicitly agent-based, showed a less-pronounced effect of the *B* variable than that seen in the later models, but the variable was nonetheless strongly predictive of outcomes. Yet Model 1 showed, across a wide parameter space, that the cost of being detected or tricked was unlikely to have an impact on the cultural transmission process. Even where *B* = 0, the value at which mimics were the least likely to effective trick receivers, increasing *cost* from 0 to 100 had little effect.

In Model 3 in particular, where we introduced a population-level effect, each 0.1 increment increase in *B* had a substantial effect on outcomes, showing an inverted r-shape. *Cost*, by contrast, was still a poor predictor, though *cost* values of 20 and 80 (where *B* = 0.6) were more likely to result in a green trait going into fixation. This result was highly significant, though the effect was not large (ORs = 1.39 and 1.36, respectively), so it is possible that with an increased number of runs, further findings might differ.

The findings from the punisher model (Model 5), however, differ marginally from those of Models 1–4: the boundaries of sensitivity to mimicry and mimicry could be larger without benefiting the bluebeard population. Also, when a value for *B* was used where no population had an advantage (0.8), *cost* values of 20 or greater benefited the greenbeard population,

though not in an increasing manner. This suggests that the addition of a punishment mechanism alters the variable values under which a greenbeard population will win out over time, even when the initial benefits to the bluebeard population of Model 4 are included. These benefits (in terms of ORs for greenbeard-trait fixation) are, however, much lower than those seen with high sensitivity B values in models 3 and 4. Punishment may therefore confer benefits over a wider array of possible scenarios, but with less effect than a simple scenario where a population of honest signallers has high average sensitivity to mimicry.

These findings have implications for researchers interested in reciprocity (direct, indirect, reputation-based, and so forth), kin selection/recognition, and greenbeard mechanisms that aim to explain the evolution of unconditional helping in various taxa. Many models, for example the classic indirect reciprocity models using image scoring or standing (for example, [44, 99]), assume perfect knowledge of others' actions (although see [100]). Yet insofar as individuals can effectively mask their cultural traits, group memberships, or even motivations, insufficient attention to detection creates a potentially serious problem for previous findings. Further, insofar as recognition of mimicry is high, relative to mimicry quality [101], the costs of detection can be low and potentially still provide strong protection against the threats that accompany mimicry (greater predation, resource exploitation, mate theft; see [102–104]). These low costs may, furthermore, make the effects of these interactions difficult to detect in natural settings. Future research, whether conceptual or empirical, could be directed to address these problems.

Finally, sociolinguistic research has for many years demonstrated how linguistic variables are associated with social group membership [105–107] and how speakers will often 'accommodate'—adapting their speech—to more closely align with their interlocutor [108, 109]. In this work, our markers of group membership are variants of a single variable: 's' and 'esh' (or green and blue in the simulations). This is a great simplification of real-world scenarios, but akin to the Shibboleth story. Accommodation theory assumes cooperative interlocutors, well disposed to each other, who are honestly co-constructing a dialogue. In a scenario like ours, where speakers from different groups aim to deceive or detect the other, our experiments suggest that the power balance is biased towards representatives of the in-group: their ability to detect imitation by members of the other group has the greater impact on eventual outcomes over generations. If the outsiders are able to evade detection, then they are able to freely exist within the in-group, passing on the mimicked trait to the next generation thereby enacting a linguistic merger of allophonic variants for their social group.

We also note that several aspects of these models may be updated to be more realistic in future study. For example, we chose only ten variables between 's' and 'esh' on a phonemic scale. In a broader study of linguistic and cultural diversity, it would be interesting to evaluate more traits as they disperse in a population. Similarly, we did not allow for mutation of fitness, cultural traits, or mimicry between populations, or for any error in detection within an individual agent's tolerance—decisions that we took for the sake of simplicity. A model that evaluates more closely mutation—and in a more complex model with multiple cultural traits, recombination—may reveal further dynamics of interest to both researchers interested in cultural evolution and sociolinguistics.

Nonetheless, from the perspective of signalling theory, these findings, within the context of evolutionary ecology and sociolinguistics, suggest that the detection of mimicry of signals of identity in intergroup conflict is likely to drive the maintenance of, or increase, signal complexity [110]. Regardless of whether the costs of being detected are high, selection may favour a high average sensitivity to mimicry within groups, such that markers of in-group affiliation are unlikely to be conventional signals, and will therefore be difficult to fake. This has implications for the selection pressures (eg, social and functional; see [84]) on linguistic variables used

within groups. Future research should focus specifically on whether the interplay between cost and detection risk affects the rate of change of these cultural traits.

## Conclusion

This series of Shibboleth models, which, to the best of our knowledge, was the first to use a variable that explicitly mirrored mimicry and sensitivity to mimicry, suggested that recognition of dishonest signalling is a better predictor of outcome than is cost. This suggests that research in theoretical biology should focus to a greater degree on detection of out-group members, be they genetically or culturally dissimilar, rather than on the costs imposed on outsiders who are detected. These findings accord with those of previous models, which suggested that mimicry is a driver of the evolution of ethnic markers and stable strategies in the evolution of cooperation [62, *inter alia*]. Future models, which rely on mutation and evolutionary game theory, may further elucidate these results. We suggest that researchers in the empirical sphere look to recognition of potential cheaters, rather than the cost of cheating only, when evaluating the impact of exploitative behaviours in social organisms.

## Supporting information

**S1 File.**
(ZIP)

## Acknowledgments

The authors are grateful to Daniel Nettle, Nik Chaudhary, and two anonymous reviewers, who gave helpful feedback on previous drafts of this article. The second author is supported by Cambridge University Press & Assessment.

## Author Contributions

**Conceptualization:** Jonathan R. Goodman.

**Formal analysis:** Jonathan R. Goodman.

**Methodology:** Jonathan R. Goodman.

**Supervision:** Andrew Caines, Robert A. Foley.

**Visualization:** Jonathan R. Goodman.

**Writing – original draft:** Jonathan R. Goodman.

**Writing – review & editing:** Jonathan R. Goodman, Andrew Caines, Robert A. Foley.

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
