## [Decision Letter · Decision Letter 0]

11 Nov 2022

PONE-D-22-24530

Shibboleth: an agent-based model of signalling mimicry

PLOS ONE

Dear Dr. Goodman,

Thank you for submitting your manuscript to PLOS ONE. After careful consideration, we feel that it has some merit but does not meet PLOS ONE’s publication criteria as it stands. Therefore, we invite you to submit a revised version of the manuscript that addresses the points raised during the review process.

We look forward to receiving your revised manuscript.

Kind regards,

Richard A Blythe

Academic Editor

PLOS ONE

Journal Requirements:

3. Please upload a copy of Supporting Information Figures 1,2,3 and 4 which you refer to in your text on page 10,20,22 and 23.

4. Please upload a copy of Supporting Information Table 1 which you refer to in your text on page 10.

Additional Editor Comments:

I apologise for the delay in a decision on this manuscript: a reviewer who promised a report was then unable to complete their assignment. The review I have received is however very thorough, and having now read the manuscript closely myself, I find myself in strong agreement with all the points they have made. Although this reviewer recommended a rejection, I think it may potentially be the case that you could address the issues in a revision. Please contact the journal office if you believe such revisions would take longer than the timescale indicated below (entered automatically by the system: I would not have chosen it).

I particularly wish to echo the reviewer's comments around the definition of the model. It is important that the main text contains sufficient details of the model that a patient reader could construct their own implementation without referring extensively to supplementary material or code written in a language that they may not use. It is unfortunate that the supplement did not reach the reviewer: I checked the file inventory and could not find it either, so I am not sure what happened here.

If you choose to resubmit, please ensure that all the points raised are addressed through revisions to the manuscript, unless there are criticisms with which you fundamentally disagree, in which case you may reply to justify your position. It is very likely that I would consult with this reviewer again on resubmission.

Reviewers' comments:

Reviewer's Responses to Questions

**Comments to the Author**

1. Is the manuscript technically sound, and do the data support the conclusions?

Reviewer #1: Partly

2. Has the statistical analysis been performed appropriately and rigorously? 

Reviewer #1: I Don't Know

3. Have the authors made all data underlying the findings in their manuscript fully available?

Reviewer #1: Yes

4. Is the manuscript presented in an intelligible fashion and written in standard English?

Reviewer #1: Yes

5. Review Comments to the Author

Reviewer #1: This paper aims to explore outgroup detection when the outgroup mimics the ingroup’s phenotype, depending on the tolerance of deviation from own phenotype and the extent of mimicry (as one variable), and the cost of outgroup detection. The literature review is well written, and overall the text as such is easy to read. However, I have several concerns regarding the model. The model assumptions are not sufficiently motivated, the model is underspecified in the manuscript, and the results are hard to interpret. Since I could not follow the model in detail, I cannot with certainty say whether it is appropriate, but from my current understanding, I think it is not only a matter of clarification, but that the model needs to be majorly redesigned to study the issue at hand.

# Model specification

It is unclear what happens in the model at many places. There is supposed to be a standard protocol in the supplement, but at least in the online supplement (there was none attached with the manuscript) there is no such protocol. Either way, the main workings of the model should be described in the main text.

I wonder whether the modelling approach is optimal. The authors have chosen to do an agent-based model. The advantages of such a model is that you can model basic mechanisms at the individual level, usually at the cost of transparency that more simple and broad-strokes models provide. However, most actions seem to concern the population level. There is heterogeneity in some parameter values and agents are paired in dyads, but the consequences concern the whole group. It seems to me that it would be possible to capture what is going on in a simpler population model. Alternatively, the model could be made truly agent-based, such that everything happens at the individual level, and if a dyadic interaction has consequences for more agents than those two involved, then that could be modelled explicitly. The model would then explain e.g. how other agents lose fitness if a certain agent fails to detect an outgroup individual, and population-level properties would emerge instead of being assumed. Currently, the model has the drawbacks of both population and agent-based models, without utilising their advantages.

Line 233: Agents "might be reproduced in groups to which their parents did not belong". How? When does this happen? (And what does it reflect in reality?) Why are they assigned a random tolerance? What are the effects for the model outcomes?

How are tolerance boundaries updated? How are traits copied?

Line 279: Explain what a "Fisher–Wright equivalent 'drift' model" means, in particular in this case.

## Minor remarks

It is a bit curious that the model is described as being specifically about first allophones on a scale and then about colours on the spectrum from blue to green, the latter even to the extent of specifying R colour packages. It is fine to provide these as examples of what the numbers representing group membership can represent, but it should be clear that the model is about numbers on an ordinal scale.

It seems the simulations are not run for sufficiently many iterations. E.g. on line 313, 14% of the runs are unresolved. Could this not be solved just by increasing the number of iterations?

# Model assumptions

There are many specific modelling assumptions that seems more or less arbitrary. Many of these are probably important drivers of the results, so it is important to understand the rationale for the assumptions, and their effects.

A central assumption is that the tolerance of deviation and potential of mimicry use the same variable. Why is this? In reality, I would rather have guessed that these would be anticorrelated. The better the other group is at mimicking my phenotype, the more sensitive I should be to accept only those that are really close to my phenotype, rather than the opposite. This also seems to be a main driver of the results. If both tolerance of deviation and potential of mimicry are small, then greenbeards will do well, and at the other end, bluebeards will win. What do we learn from this?

Another coupling is that the cost when a greenbeard accepts a bluebeard is the same as it is for a bluebeard to be detected. What is the rationale for this? I would expect greenbeards to have to pay to sanction the bluebeard, but, on the contrary, now the cost comes when they do not sanction. Given the symmetry in costs, it seems just to moderate the size of the effect when bluebeards are or are not detected, and it comes as no surprise that varying the cost variable does not change the results qualitatively. But does this really model cost of sanctioning?

Line 221: "Where bluebeards make up less than half of the population, boundaries increases the global tolerance boundaries range by (traits-1)*B". Why? What is the rationale for this and why is there a sharp cut-off? Why is this directional? And how is it implemented? This might have large consequences.

Line 230: Where does the extra fitness cost for being a discerning receiver or more precise mimic come from (in reality)? Could this be modelled specifically instead of as a general cost?

Figure 2: Why does everyone in the group lose PTR when one individual accepts a bluebeard, or a bluebeard is detected? This supposedly has far-reaching consequences for the outcomes. Why does every individual agent pay a tolerance score cost when a bluebeard has passed the mimicry test? Either the score is a cost for the agent paired with the bluebeard agents, but then only that individual should pay, or it is an investment cost to have such a detection system, but then that cost should not be contingent on a bluebeard agent having passed the test (with another agent).

Model 2: Why is it interesting to study the case where agents have a higher chance of adopting traits 6 and 7? Why are they specifically five times more likely to adopt those traits? Why do they pay twice as high a cost (i.e. why is it higher, and why specifically twice)? And why are both changes made to the model at once?

Model 3: "a low-tolerance greenbeard paid a fitness cost to reduce the high-tolerance greenbeard's tolerance to 0" and "Cost was twice the cost variable". Why would anyone do this in reality? Why is it reduced all the way to 0? And how is the “altruistic punishment” cost decided?

## Minor remarks

The authors "remain agnostic as to whether the traits are cultural or genetic", but this should have some consequences for the model, e.g. as to whether agents can change their traits.

Why are "boundaries" proportional to "traits-1" instead of just "traits"?

Figure 2: Should it be cost/10-tolerance, instead of cost/10+tolerance? The cost increases the less tolerant the agent is. And why is cost divided by 10, instead of just using values of the parameter that are ten times smaller than now?

# Model interpretation

Related to the assumptions, it is often hard to interpret what the results of the model mean. Overall, I wonder if the right thing is being measured.

I wonder if it is correct to see the greenbeard phenomenon as a situation of competition between groups. You want to exclude freeriders from benefitting from cooperative norms in your group, not to compete with another group. It is not modelled here what is the difference between the greenbeards and the bluebeards, only that bluebeards want to mimic the greenbeards. It seems in the model that agents (or their offspring) can change groups. What does it mean then, when a group has gone into fixation? If everyone has adopted a green trait, is that really a success for the greenbeards, if the purpose is usually to keep the others out? Most of the analyses are about which group has taken over the entire population, but I wonder if this is really the right measure, and how to interpret such a result. Had the model been explicit about what is at play, and modelled within-group interactions explicitly, then it would have been possible to measure e.g. whether cooperation is maintained within the greenbeards, or if benefits from cooperation outweigh costs of trying to keep bluebeards out (either through detection or through imposing higher costs), which is what I think the authors are actually after.

A main result is that detection is more important than costs. But this seems to be fundamentally built into the assumptions, as described above. Acceptance is coupled positively(!) with the potential of the outgroup to mimic, and costs are applied equally to both sides. I would be hesitant to make interpretations of this to apply to detection and sanctioning frauds in general. Line 462: “we found that B … was an excellent predictor”, but it is designed so that low values favour green and high values favour blue.

## Minor remarks

Some tables and figures are missing, or have the wrong numbers, e.g. Figure S1 (or Figure 7 on line 456 or Figure 6 on line 466) is not what is decribed on line 168, and there are no tables in the supplement (or a Figure 19, line 550).

There are many figures that seems to say very similar things. I think the number of figures could be reduced considerably and be more to the point.

# Literature review

As I said above, the literature review is generally well written, pertaining to the background on groups and cooperation. However, there should probably be more focus on references to explain and motivate the model assumptions. I also have a few suggestions for references that seem relevant for the background:

## Parochial altruism and tag-based cooperation

- Fu, F., Tarnita, C. E., Christakis, N. A., Wang, L., Rand, D. G., & Nowak, M. A. (2012). Evolution of in-group favoritism. Scientific Reports, 2, 1–6. https://doi.org/10.1038/srep00460

- Jansson, F. (2015). What games support the evolution of an ingroup bias? Journal of Theoretical Biology, 373, 100–110. https://doi.org/10.1016/j.jtbi.2015.03.008

## Greenbeards and language

- Lindenfors, P. (2013). The green beards of language. Ecology and Evolution, 3(4), 1104–1112. https://doi.org/10.1002/ECE3.506

- Jansen, V. A. A., & van Baalen, M. (2006). Altruism through beard chromodynamics. Nature, 440(7084), 663–666. https://doi.org/10.1038/nature04387

- Cohen, E. (2012). The Evolution of Tag-Based Cooperation in Humans. Current Anthropology, 53(5), 588–616. https://doi.org/10.1086/667654

## Population structure

I do not think that not assuming a lattice structure is something that needs to be motivated. There are problems with such assumptions, and alternative approaches might be better, as illustrated in these papers, in case the authors still want to keep such a motivation in the manuscript:

- Jansson, F. (2013). Pitfalls in Spatial Modelling of Ethnocentrism: A Simulation Analysis of the Model of Hammond and Axelrod. Journal of Artificial Societies and Social Simulation, 16(3), 2. http://jasss.soc.surrey.ac.uk/16/3/2.html

- Read, D. (2010). Agent-based and multi-agent simulations: coming of age or in search of an identity? Computational and Mathematical Organization Theory, 16(4), 329–347. https://doi.org/10.1007/s10588-010-9067-1

- Tarnita, C. E., Antal, T., Ohtsuki, H., & Nowak, M. A. (2009). Evolutionary dynamics in set structured populations. PNAS, 106(21), 8601–8604. https://doi.org/10.1073/pnas.0903019106

6. PLOS authors have the option to publish the peer review history of their article (what does this mean?). If published, this will include your full peer review and any attached files.

Reviewer #1: No

---

## [Author Response · Author response to Decision Letter 0]

20 Mar 2023

Please see our 'response to reviewer' file included in this submission.

---

## [Editor Report · Decision Letter 1]

30 Mar 2023

PONE-D-22-24530R1Shibboleth: an agent-based model of signalling mimicryPLOS ONE

Dear Dr. Goodman,

Thank you for resubmitting your manuscript to PLOS ONE.

I intend to send this resubmission out for a second round of review. However, there are some technical problems with the PDF which I would like you to resolve before I do so.

First, the figures do not display properly in the PDF - pages 56-66 each contain an error message, stating that the image uploaded is corrupt or invalid. Please could you take a look again at your figures and ensure that they appear correctly, subsequently to the manuscript, before finalising the submission.

Second, your response to the reviewer includes Word comments in the margin. One of these has remained in the version that has been submitted, which I presume was intended as private communication between authors rather than something you wanted the reviewer to see. Please could you excise this comment in the resubmission.

If you run into any problems, please contact the journal office at plosone@plos.org for assistance.

We look forward to receiving your revised manuscript.

Kind regards,

Richard A Blythe

Academic Editor

PLOS ONE
---

## [Author Response · Author response to Decision Letter 1]

31 Mar 2023

Dear Dr Blythe,

Thank you very much for sending us the reviewer’s comments, which we have found extremely helpful. And we apologize for the delay in sending our response — the edits required have been both computationally demanding and time consuming.

We would like to respond to each of the reviewer’s concerns in turn, and have given their comments in italics followed by our response in bold in a response file, and made changes where relevant in the manuscript and supplement.

Best wishes,

Jonathan R Goodman

---

## [Decision Letter · Decision Letter 2]

8 Jun 2023

PONE-D-22-24530R2Shibboleth: an agent-based model of signalling mimicryPLOS ONE

Dear Dr. Goodman,

Thank you for submitting your manuscript to PLOS ONE and your patience with us during the review process. The reviewer of your original submission was unfortunately unavailable, but I managed to find another reviewer who has taken a look at the manuscript in the light of the original reviewer's comments and your response. This reviewer has a generally favourable opinion of the manuscript, but suggests some clarifications which I think are worthy of consideration before making a final decision on the manuscript. Therefore, we invite you to submit a revised version of the manuscript that addresses the points that have been raised. My hope is that I will be able to make a final decision without the need for further review. My impression is that the comments can mostly be addressed without undue amounts of extra work: where this is evidently not the case, it might be acceptable to acknowledge the point as one for a future study.

We look forward to receiving your revised manuscript.

Kind regards,

Richard A Blythe

Academic Editor

PLOS ONE

Journal Requirements:

Reviewers' comments:

Reviewer's Responses to Questions

**Comments to the Author**

1. If the authors have adequately addressed your comments raised in a previous round of review and you feel that this manuscript is now acceptable for publication, you may indicate that here to bypass the “Comments to the Author” section, enter your conflict of interest statement in the “Confidential to Editor” section, and submit your "Accept" recommendation.

Reviewer #2: (No Response)

2. Is the manuscript technically sound, and do the data support the conclusions?

Reviewer #2: Partly

3. Has the statistical analysis been performed appropriately and rigorously? 

Reviewer #2: N/A

4. Have the authors made all data underlying the findings in their manuscript fully available?

Reviewer #2: Yes

5. Is the manuscript presented in an intelligible fashion and written in standard English?

Reviewer #2: Yes

6. Review Comments to the Author

Reviewer #2: Summary

What maintains linguistic diversity? This theoretical study investigates one mechanism, which is the association of particular linguistic variants with parochial altruism. This is in part inspired by the Biblical story of Shibboleth, where a particular pronunciation of the word Shibboleth is used as a marker of in-group, out-group status. The authors construct an agent based simulation to consider the scenario when two populations meet (“Greenbeards” and “Bluebeards”), and consider how differing degrees of tolerance, mimicry, as well as costs and benefits to these actions (as well as failure to detect or mimic) may alter the effects of parochial altruism on the frequency of different linguistic phenotypes.

I think the question that the authors investigate is an interesting one, although I have questions about why they chose the particular modelling approach that they did (with many of the same queries as the previous reviewer), where perhaps a simpler modelling approach would have been clearer to understand and generated many of the same behaviours. For example, why simply not simply two variants with probabilities of acceptance/rejection (rather than the 10 variants that are used), or why not have a symmetrical interaction structure where individuals can interact with anyone in the population, then you would be able to investigate potentially other types of signalling error (i.e. rejecting someone who is actually of the same type). I think some of this would appear more natural (at least to me!) than the setup investigated.

Nonetheless, the authors have responded in a substantial manner to the previous review, and so I think many of the issues I have can be dealt with by a simple clarification of the methods. I also think the authors have done an excellent job of making their code associated with their simulations accessible, so that others in the future will also be able to probe and build upon their models in the future.

Main points

The major point that I still do not think is very clear is the underlying demography and inheritance structure. As far I can see we begin with an equal number of “Greenbeards” and “Bluebeards”, and then individuals reproduce according to their fitnesses (which arise from a combination of the costs and benefits they experience during the social interactions). This presumably means that it likely that in the next generation there will be an unequal number of “Greenbeards” and “Bluebeards”. Does this that the less frequent groups have individuals that pay twice the penalty/get twice the benefits from the social interactions? Or does it mean certain individuals do not undergo the social interaction? For example, what does it mean “Where traits were reproduced into the group to which their cultural parents did not belong (L255-256)”. I don’t really think this aspect is described, and so again I think more explanation is required to outline how this aspect of the demography functions.

Related to this it is unclear to me both some of inheritance of certain traits, or why certain choices about what is heritable/what is not were made. This is important, because of course the choices about this govern what aspects of the system are evolving. It appears that individuals simply inherit the trait of their parent with perfect fidelity. Is this correct? If so, that presumably means the only source of linguistic variation is that which the population started with (i.e. there is no equivalent of mutation in the model). Similarly, I was confused about what traits were heritable (and therefore potentially subject to evolutionary pressures in the model). For example, parent “fitness” seems to be. Does this mean that individuals themselves do not start form a blank slate, but instead inherit the fitness state of their parents? If so, this seems an unconventional choice, and I am not sure what it adds to the model. Similarly, it seems like mimicry was heritable. So again, we would expect the mimicry values to change throughout the course of the model, but as far as I can see this is not commented on? Again, it seems odd to assume mimicry is heritable but tolerance is not (except in Model 5). Again here the evolutionary dynamics of toleration is not commented upon (as far as I can see). [Indeed, I think the authors could make clearer from the jump what aspects of their system they are investigating the evolution of, and which are simply random variables each generation.]

The central finding, which the authors focus on, is the fact that B, which then scales the range outside one’s focal value with which one is tolerant, and scales the range outside which one can mimic, has a strong effect on whether the detector/mimic increases in frequency. Maybe unsurprisingly, when individuals can mimic a greater range of values they increase in frequency (“Bluebeards” more likely to go fixation), and when they cannot they don’t (“Greenbeards” more likely to go fixation). They find that this pattern holds across many of their simulations. As they point out in their previous response to reviewers this might seem “trivial”, although I don’t see that as an issue, one would hope that behaviours of a model that one expects to see do actually emerge in the outcomes of the model. Moreover, one might expect it to interact more strongly with costs, although this is hard to see outside of Model 1 as they typically do not vary both factors simultaneously.

In contrast to B, across their models, they find that the absolute various costs imposed have little qualitative effect, although as their model 2 shows, potential asymmetries in costs may change the outcome qualitatively.

As an aside, it is a shame that for all the runs the authors weren’t able to keep the simulation going until fixation. Because one of the interesting aspects of the model would be not only to look at whether green/blue was more likely to fix, but the average time to fixation. This would be a nice proxy for how well this particular system might be able to maintain heterogeneity of signals. The authors do mention that in their Model 1 fewer runs went to fixation than in the drift scenario, and this would be an interesting angle to develop, and I wonder whether they noticed any other patterns determining the proportion of runs that went to fixation in the population. Alternatively, they could look at another measure of signal diversity in their populations through time (maybe they could look at something equivalent to heterozygosity) and investigate the rate at which it declines. Maybe here factors like costs do have an impact affect outcome.

Finally, I do wonder whether to streamline the paper, more of the focus could be placed on Model 1, as this is the simplest, most thoroughly investigated, and captures much of the behaviour that the authors discuss, with the other models made a little more ancillary (or even moved into the Supplementary Material). But this is purely a personal preference.

Minor points

The referencing between colours/phonemes feels a bit unnecessary. Ultimately, these are all just illustrations of it being a number in the actual mechanics of the model. Whilst having the illustration helps, flitting between them or adding in oddly specific details (e.g. “Specifically, we assume for convenience that there are ten linguistic variables between esh (ʃ; voiceless palato-alveolar fricative) and s (unvoiced alveolar sibilant), which we represent on an ordinal scale with a range of 1-10, and visualise on a colour spectrum between green (esh) and blue (s).”) which frankly have little to do with the actual model they have created and are just a little distracting. I would suggest slimming down and focusing on the colour element (although this is more a personal choice of what I think is easier for the reader to follow).

Many of the parameters in the model, do seem chosen a little arbitrarily, and it’s not clear what effect they have on the results. For example, in the population cost model everyone in the population pays the costs for the success/failure of the interaction, why was 10 chosen as the factor? Or indeed why were 10 phonemes chosen in the first place? I understand that with a model such as this there will be aspects that simply need to be chosen, but then I think this speaks to the strengths of a simpler model, with fewer parameters, whose behaviour can be more thoroughly investigated.

L22 - “Here we describe the results of three agent-based models that simulate multi-generational interactions between two groups of individuals” – I thought it was 5?

7. PLOS authors have the option to publish the peer review history of their article (what does this mean?). If published, this will include your full peer review and any attached files.

Reviewer #2: No

---

## [Author Response · Author response to Decision Letter 2]

30 Jun 2023

Dear Dr Blythe, 

Thank you very much for reconsidering this submission, and for sending the second reviewer's comments, which we found extremely useful for further revision. We attach in this submission a response to each of the reviewer's comments.

With best wishes,

Jonathan

---

## [Editor Report · Decision Letter 3]

18 Jul 2023

Shibboleth: an agent-based model of signalling mimicry

PONE-D-22-24530R3

Dear Dr. Goodman,

Thank you for resubmitting your manuscript to PLOS ONE, and for your consideration of the reviewer's comments. I have read through the manuscript, and find the revisions satisfactory. This means your manuscript is now judged scientifically suitable for publication and will be formally accepted for publication once it meets all outstanding technical requirements.

Kind regards,

Richard A Blythe

Academic Editor

PLOS ONE
---

## [Editor Report · Acceptance letter]

20 Jul 2023

PONE-D-22-24530R3 

Shibboleth: an agent-based model of signalling mimicry 

Dear Dr. Goodman:

I'm pleased to inform you that your manuscript has been deemed suitable for publication in PLOS ONE. Congratulations! Your manuscript is now with our production department. 

Kind regards, 

on behalf of

Prof. Richard A Blythe 

Academic Editor

PLOS ONE